# Low Incidence of Hepatocellular Carcinoma after Antiviral Therapy in Patients with Chronic Hepatitis C and Hemophilia

**DOI:** 10.3390/jcm11051451

**Published:** 2022-03-07

**Authors:** In Jung Kim, Sung Hwan Yoo, Sora Kim, Young Youn Cho, Ki Young Yoo, Hyung Joon Kim, Hyun Woong Lee

**Affiliations:** 1Department of Internal Medicine, Gangnam Severance Hospital, Yonsei University College of Medicine, Seoul 06273, Korea; sginjung@naver.com (I.J.K.); oscar0125@yuhs.ac (S.H.Y.); lobvbol13@yuhs.ac (S.K.); 2Department of Internal Medicine, Chung-Ang University College of Medicine, Seoul 06973, Korea; mdcho@cau.ac.kr; 3Department of Pediatrics, Korea Hemophilia Foundation Hospital, Seoul 06641, Korea; gowho@hanmail.net

**Keywords:** chronic hepatitis C, hemophilia, hepatocellular carcinoma

## Abstract

Background: Hepatocellular carcinoma (HCC) rarely develops in patients with chronic hepatitis C (CHC) who achieve sustained virological response (SVR). We assessed the incidence of HCC in CHC patients with hemophilia after treatment with pegylated interferon plus ribavirin (PegIFN/RBV) and direct-acting antivirals (DAAs). Methods: Patients (*n* = 202) were enrolled between March 2007 and July 2019. A total of 139 patients were treated with PegIFN/RBV (genotype 1, *n* = 98; genotype 2, *n* = 41). Sixty-three patients were treated with DAAs (genotype 1, *n* = 44; genotype 2, *n* = 19). The cumulative incidence rates of HCC were estimated using the Kaplan–Meier method and compared using the log-rank test. Results: For genotype 1, SVR was achieved in 78.6% (77/98) and 90.9% (40/44) of patients in the PegIFN/RBV and DAAs groups, respectively. For genotype 2, SVR was achieved in 95.1% (39/41) and 94.7% (18/19) of patients in the PegIFN/RBV and DAAs groups, respectively. Six HCC cases were identified. The cumulative incidence of HCC was 4.1% at 14 years in PegIFN/RBV and 1.7% at 5 years in DAAs. The 14-year cumulative incidence of HCC was 1.9% in the SVR group and 21.7% in the no-SVR group in the PegIFN/RBV group (*p* < 0.001). Conclusions: Treatment with PegIFN/RBV led to stable SVR and a low incidence of HCC. Although the follow-up period was short, DAAs led to more stable SVR than PegIFN/RBV and a low incidence of HCC in CHC patients with hemophilia.

## 1. Introduction

Hepatitis C virus (HCV) infection causes chronic hepatitis C (CHC), which is the leading cause of liver cirrhosis (LC) and hepatocellular carcinoma (HCC) [1,2]. Interferon therapy has been used for treating HCV since the 1980s. However, the variance in interferon treatment success rates due to differences in HCV genotype, human immunodeficiency virus (HIV) coinfection, and liver fibrosis has emerged as a problem [3,4,5]. Interferon therapy can cause multiple side effects and worsen patient compliance, resulting in a high rate of treatment discontinuation and failure [6]. Since 2011, direct-acting antivirals (DAAs) have enabled advances greatly in the treatment of HCV through high-sustained virological response (SVR) rates, high barrier to viral resistance, and treatment for all genotypes [7,8,9]. In the early 1990s, most patients with hemophilia were treated with contaminated blood-derived clotting factor products. They were infected with HCV, and 80% of them had CHC [10,11]. Untreated CHC can progress to liver fibrosis and cirrhosis, which can ultimately cause decompensated LC, HCC, and death [12]. Over time, treatments for patients with hemophilia have improved remarkably and life expectancy has increased [13]. Therefore, the management of HCV infection and control of various complications are emerging as important factors to improve the long-term prognosis of patients with hemophilia [13]. Our team demonstrated that pegylated interferon plus ribavirin (PegIFN/RBV) and DAAs are effective in the treatment of CHC patients with hemophilia [14,15]. In previous multicenter study, DAAs were safe and highly effective in hemophilia with HCV infection [16]. Recent studies have demonstrated that HCC occurs in patients with CHC who have achieved SVR, although the frequency is low. In addition, it has been reported that HCC also occurs in patients using DAAs [17,18]. However, there are no studies on the long-term treatment of HCC in patients with inherited bleeding disorders and CHC.

Therefore, this study aimed to assess the incidence of HCC after PegIFN/RBV and DAA therapy in HCV-infected patients with hemophilia. 

## 2. Materials and Methods

### 2.1. Study Population

Since 1991, the Korea Hemophilia Foundation (KHF) has been treating patients with hemophilia and improving their quality of life. We obtained data from 202 patients with hemophilia and HCV who visited the KHF between March 2007 and July 2019 (Figure 1). A total of 2117 patients with hemophilia underwent anti-HCV antibody testing. Of these patients, 575 underwent HCV RNA testing. HCV RNA was positive in 258 patients (44.9%) and negative in 317 patients. All 258 patients maintained HCV RNA positive for more than 6 months. We genotyped 235 patients. Hepatitis C virus genotyping was carried out using the restriction fragment mass polymorphism method. Of the 235 patients who tested for HCV genotype, 202 patients (86.0%) were treated for CHC, and 33 patients were untreated. The number of patients with genotype 1 was 98 (70.5%) and 44 (69.8%) in the PegIFN/RBV and DAA groups, respectively. PegIFN/RBV was treated for 48 weeks and DAA was treated for 8–12 weeks. The number of patients with genotype 2 was 41 (29.5%) and 19 (30.2%) in the PegIFN/RBV and DAA groups, respectively. PegIFN/RBV was treated for 24 weeks and DAA was treated for 8–12 weeks. There was no patient who used the combination of PegIFN/RBV and DDA in this study. This combination therapy had not been supported by the medical insurance system in Korea.

All patients were aged >20 years. Patients with HBV or HIV coinfection were also included in this study. Additionally, requirements included compensated liver disease with a prolonged prothrombin time of <4 s, serum albumin ≥3.0 g/dL, total bilirubin ≤4 mg/dL, and no history of hepatic encephalopathy or bleeding esophageal varices. We diagnosed cirrhosis with ultrasonography and excluded candidates who had a history of malignancy, including HCC, or a history of intravenous drug abuse. Patients with other significant psychiatric problems were excluded.

The study was performed in accordance with the Declaration of Helsinki (1975) and was approved by the Institutional Review Board of Gangnam Severance Hospital (3-2020-0166). As this study was retrospective in nature, written consent from the patients was not required. 

### 2.2. Approved DAAs

A number of DAAs have been approved for the treatment of hepatitis C. Among them, six DAAs have been available in patients with chronic hepatitis C and hemophilia in Korea. DAAs are as follows. 1. Daklinza^®^ (daclatasvir) plus Sunvepra^®^ (asunaprevir), 2. Sovaldi^®^ (sofosbuvir) plus ribavirin, 3. Harvoni^®^ (ledipasvir/sofosbuvir), 4. Zepatier^®^ (elbasvir/grazoprevir), 5. Viekira Pak^®^ (dasabuvir plus ombitasvir-paritaprevir-ritonavir), 6. Maviret^®^ (glecaprevir/pibrentasvir).

### 2.3. Assessment of Efficacy

The primary endpoint was the cumulative incidence of HCC. The secondary endpoint was SVR, defined as the absence of detectable HCV RNA (lower limit of detection, ≤15 IU/mL) at 12 weeks after the end of therapy, using real-time HCV RNA quantification (Roche Diagnostics, Branchburg, NJ, USA). Regardless of the presence of SVR, all patients should undergo surveillance for HCC every 6 months by means of ultrasound and alpha-fetoprotein. Patients with SVR undergo HCV RNA test every 6 or 12 months to evaluate relapse or reinfection.

### 2.4. Statistical Analyses

The baseline characteristics were expressed by Student’s *t*-test for continuous variables and the chi-square test for categorical variables. All data are expressed as the mean ± standard deviation or range, where appropriate. We used the Kaplan–Meier plot to estimate the cumulative incidence rates of HCC and the log-rank test to assess whether the curves showing the development of HCC were significantly different in patients who achieved SVR with those who did not. All statistical analyses were performed using SPSS software (version 25.0; IBM Corp., Chicago, IL, USA). Statistical significance was set at *p* < 0.05.

## 3. Results

### 3.1. Baseline Characteristics

Baseline characteristics of the study population are presented in Table 1. The study population comprised 202 patients, including 139 in the PegIFN/RBV group and 63 in the DAA group. The mean age was 48 years (range, 33–83 years) for genotype 1 and 48 years (range, 34–83 years) for genotype 2. Of the 202 participants, 16 patients (7.9%) had compensated liver cirrhosis, including 11 (six on PegIFN/RBV and five on DAAs) with genotype 1 and 5 (three on PegIFN/RBV and two on DAAs) with genotype 2, respectively. For genotype 1, the mean baseline serum alanine aminotransferase (ALT) levels were 65 IU/L (range, 38–255) and 66 IU/L (range, 37–238) in the PegIFN/RBV and DAA groups, respectively. For genotype 2, the mean baseline serum ALT levels were 78 IU/L (range, 30–377) and 68 IU/L (range, 28–377) in the PegIFN/RBV and DAA groups, respectively. There was no significant difference between the two groups. For genotype 1, the mean baseline serum HCV RNA levels were 6.5 log_10_ IU/mL (range, 4.1–8.2) and 6.3 log_10_ IU/mL (range, 3.7–7.1) in the PegIFN/RBV and DAA groups, respectively. For genotype 2, the mean baseline serum HCV RNA levels were 5.9 log_10_ IU/mL (range, 3.7–7.6) and 5.6 log_10_ IU/mL (range, 3.7–7.6) in the PegIFN/RBV and DAA groups, respectively. There was no difference between the two groups. Nine patients had HBV coinfection (six in the PegIFN/RBV group and three in the DAA group). One patient had HIV coinfection. (PegIFN/RBV group). The mean follow-up periods were 12 years and 3.8 years for patients treated with PegIFN/RBV and DAAs, respectively.

### 3.2. Antiviral Response

Figure 2 shows the treatment response for PegIFN/RBV and DAAs in each group. For genotype 1, SVR was achieved in 78.6% (77/98) and 90.9% (40/44) of patients in the PegIFN/RBV and DAA groups, respectively. For genotype 2, SVR was achieved in 95.1% (39/41) and 94.7% (18/19) of patients in the PegIFN/RBV and DAA groups, respectively.

### 3.3. HCC Incidence

HCC occurred in 3.6% (5/139) and 1.6% (1/63) of patients treated with PegIFN/RBV and DAAs, respectively. The cumulative incidence of HCC was 4.1% at 14 years in the PegIFN/RBV group and 1.7% at 5 years in the DAA group (Figure 3). The baseline characteristics of patients with HCC are shown in Table 2. The mean age was 77 years (range 66–83 years), five patients had genotype 1 (genotype 1a: 1, genotype 1b: 4), four patients had liver cirrhosis, and three patients (PegIFN/RBV: 2, DAAs: 1) experienced SVR (Table 2). One patient (genotype 1b) who achieved SVR on DAAs had HCC at 2 years after the end of the treatment. The 14-year cumulative incidence of HCC in patients treated with PegIFN/RBV was 1.9% in the SVR group and 21.7% in the no-SVR group (*p* < 0.001) (Figure 4).

## 4. Discussion

Our results indicate that the SVR rates of the PegIFN/RBV and DAA groups were similar to the SVR of CHC patients without hemophilia. In our study, DAAs were also effective in achieving a high SVR rate similar to that in patients without hemophilia [3,12,19]. Four of the five patients with treatment failure on DAAs received daclatasvir/asunaprevir during 2015–2016, which is thought to be the cause of treatment failure. Daclatasvir/asunaprevir is a first-generation all-oral DAA regimen, which is rarely used in current practice, and is known to have a higher resistance and lower efficacy than other DAAs [20,21,22].

Our study demonstrated that genotype 1 was predominant (139/202, 68.8%). The genotype distribution was different from that of Korean CHC patients without hemophilia. There were regional variations in the genotype ratios of the patients. Kim et al. reported that the proportion of genotypes 1 (49.2%) and 2 (50.8%) was almost the same in Korea [23,24]. We assumed that the clotting factors of patients with genotype 1 were collected when the first derived clotting factor was made. 

In a previous study, old age, male sex, higher body mass index, cirrhosis, and lower platelet count were associated with increased risk of HCC. Although the low incidence of HCC and small number of patients made it difficult to evaluate the risk factors for HCC, the mean age of the patients with HCC was significantly higher than that of the patients without HCC in this study. It is also well known that the incidence of HCC increases rapidly in older HCV patients without hemophilia [25]. They have longer follow-up periods after administration of first derived clotting factor than hemophilia patients without HCC. This means that the duration of CHC infection is longer than that in other patients without HCC. 

According to several studies, the incidence of HCC in CHC patients without inherited bleeding disorders is 1–23%. In some studies, 6–8% of patients with liver cirrhosis and HCV have HCC every year [10,26,27]. Recently, Janjua el al. reported that the cumulative incidence of HCC at 10 years was 1.06% in the SVR group and 6.47% in the no SVR group [18].

Our study demonstrated that the cumulative incidence of HCC at 10 years was 0.9% in the SVR group and 21.7% in the no SVR group. The incidence of HCC in the SVR group was similar between our study and that in a previous report. However, the incidence of HCC in the no SVR group was higher than that in patients without inherited bleeding disorder. We assume that these results are associated with the baseline characteristics of patients with CHC. The presence of liver cirrhosis in our study was higher than that reported in a previous study (7.9% vs. 5.0%). Unfortunately, our study did not analyze the incidence of HCC in hemophilia patients with CHC not receiving HCV treatment. In previous study, the incidence of HCC in hemophilia patients with CHC not receiving HCV treatment was higher than in patients with receiving successful HCV treatment (3% vs. 0.5%) [12].

HCV infection remains stable for a long period without notable symptoms, and is often found by incidental blood tests or after it has already progressed significantly. In most patients with hemophilia, the time of administration of the first clotting factor concentrates is assumed to be the beginning of HCV exposure. It is possible to control liver disease early through periodic testing and management of HCV infection [10,12]. Our study also has the advantage of analyzing the long-term effect of CHC treatment. This study was the first to investigate the incidence of HCC in hemophilia patients with CHC. In patients with HCV, such as intravenous drug abusers, it is difficult to determine the route and duration of HCV infection. In addition, they may have poor treatment compliance. In contrast, patients with hemophilia have to visit hospitals regularly, making it easy to study the natural history of CHC and effects of antiviral treatment on the disease. Although the frequency was low, HCC occurred even though patients with CHC achieved SVR. In addition, it has been reported that HCC rarely occurs in patients using DAAs [17,18]. Our study demonstrated that the incidence of HCC was very low in hemophilia patients with SVR, and the incidence of HCC was extremely low even after using DAAs. Nevertheless, HCC occurred in one patient even after SVR was achieved.

Our study had some limitations. First, even though there have been developments in treatment for hemophilia, patients are more likely to die from complications associated with bleeding (hemorrhage and trauma-induced injury), other malignancies, and HIV [28]. In our study, three patients with HCC died of brain hemorrhage, pneumonia, and leukemia. The risk of having these complications might be a limitation in the study of the long-term effectiveness of PegIFN/RBV and DAAs, or the incidence of HCC. Second, this was a retrospective study and the number of patients was small. The follow-up period for patients using DAAs was short. However, this cohort is meaningful because of long-term treatment and monitoring of patients with CHC accompanied by hemophilia.

## 5. Conclusions

PegIFN/RBV lead to stable SVR and a low incidence of HCC. Although the follow-up period was short, oral DAA treatment leads to a more stable SVR than PegIFN/RBV and a low incidence of HCC in CHC patients with hemophilia.

## Figures and Tables

**Figure 1 jcm-11-01451-f001:**
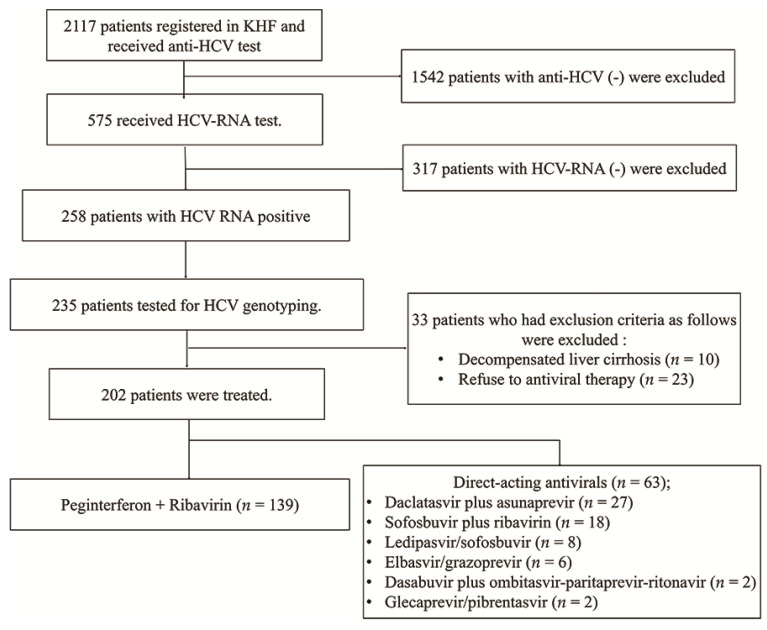
Flow and exclusion of study participants. KHF, The Korea Hemophilia Foundation; HCV, hepatitis C virus.

**Figure 2 jcm-11-01451-f002:**
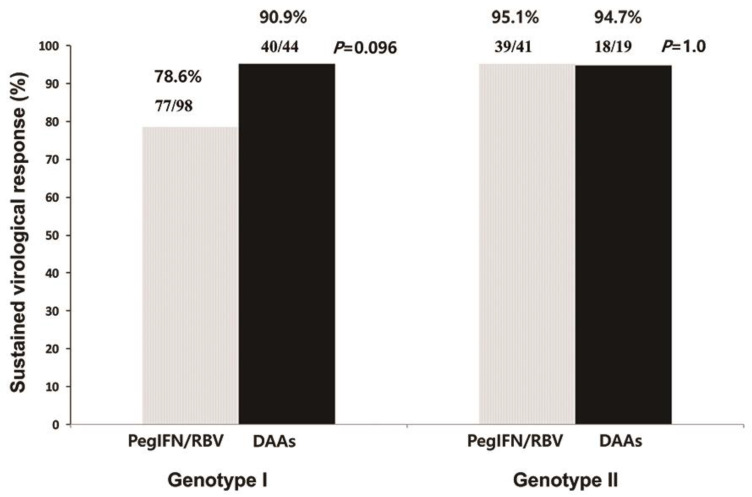
Treatment responses to pegylated interferon plus ribavirin and direct-acting antivirals. PegIFN/RBV, pegylated interferon plus ribavirin; DAAs, direct-acting antivirals.

**Figure 3 jcm-11-01451-f003:**
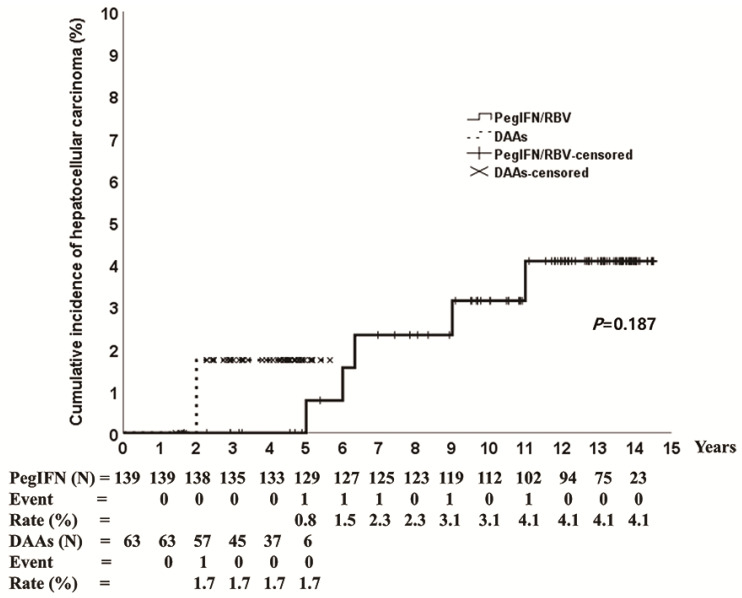
Cumulative incidence of hepatocellular carcinoma according to pegylated interferon plus ribavirin and direct-acting antivirals. PegIFN/RBV, pegylated interferon plus ribavirin; DAAs, direct-acting antivirals.

**Figure 4 jcm-11-01451-f004:**
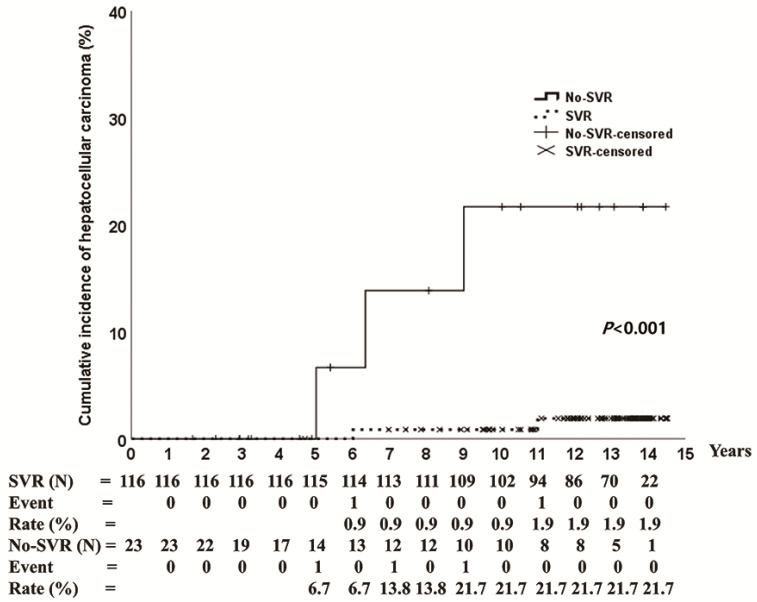
Cumulative incidence of hepatocellular carcinoma according to treatment response with pegylated interferon plus ribavirin. SVR, sustained virological response; No-SVR, no sustained virological response.

**Table 1 jcm-11-01451-t001:** Baseline clinical characteristics of patients with chronic hepatitis C and hemophilia.

Variables	Genotype 1 (*n* = 142)		*p* Value	Genotype 2 (*n* = 60)		*p* Value
	PEG-IFN (*n* = 98)	DAA (*n* = 44)		PEG-IFN (*n* = 41)	DAA (*n* = 19)	
Male (%)	98 (100)	44 (100)		41 (100)	19 (100)	
Mean age, year (range)	48 (33–83)	50 (33–79)	0.307	47 (34–83)	51 (34–83)	0.119
BMI, kg/m^2^ (range)	22.9 (20.6–41.6)	23.1 (18.7–31.6)	0.750	23.4 (16.4–34.5)	22.0 (16.9–29.7)	0.233
ALT, IU/L (range)	65 (38–255)	66 (37–238)	0.905	78 (30–377)	68 (28–377)	0.648
Platelet, ×10^3^/mm^3^ (range)	242 (98–428)	223 (98–404)	0.110	252 (125–473)	256 (140–385)	0.372
HCV RNA, log_10_ IU/mL (range)	6.5 (4.1–8.2)	6.3 (3.7–7.1)	0.174	5.9 (3.7–7.6)	5.6 (3.7–7.6)	0.824
Liver cirrhosis (%)	6 (6.1)	5 (11.4)	0.280	3 (7.3)	2 (10.5)	0.648
HIV co-infection (%)	1 (1.0)	0 (2.3)	1.000	0	0	-
HBV co-infection (%)	4 (4.1)	3 (6.8)	0.677	2 (4.9)	0 (0.0)	1.000
Follow-up period, month (range)	141 (20–174)	47 (19–68)	<0.001	152 (35–174)	45 (17–59)	<0.001

Abbreviations: PegIFN/RBV pegylated interferon plus ribavirin; DAAs direct-acting antivirals; BMI: body mass index; ALT: alanine aminotransferase; HCV: hepatitis C virus; HIV: human immunodeficiency virus; HBV: hepatitis B virus.

**Table 2 jcm-11-01451-t002:** Clinical characteristics of patients with newly developed hepatocellular carcinoma after antiviral therapy.

Patient	Drug	Age, Years	BMI, kg/m^2^	Genotype	HCV RNA, log_10_ IU/mL	ALT, IU/L	Cirrhosis	Duration *	SVR
1	PegIFN/RBV	66	22.9	1a	7.4	69	1	72	1
2	PegIFN/RBV	83	17.2	1b	7.1	88	0	108	0
3	PegIFN/RBV	74	22.6	1b	6.5	104	0	76	0
4	PegIFN/RBV	83	22.0	2a, 2c	5.0	248	1	132	1
5	PegIFN/RBV	79	22.5	1b	6.3	32	1	60	0
6	DAA ^†^	79	23.1	1b	6.2	35	1	24	1

* Duration from PegIFN/RBV or DAA therapy to the occurrence of HCC. ^†^ One patient (genotype 1b) who achieved SVR on DAAs experienced HCC at 2 years after the end of treatment. Abbreviations: PegIFN/RBV pegylated interferon plus ribavirin; DAAs direct-acting antivirals; BMI, body mass index; ALT, alanine aminotransferase; SVR, sustained virologic response.

## Data Availability

The data presented in this study are available on request from the corresponding author. The data are not publicly available due to privacy.

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
