# Peer review of "Low Incidence of Hepatocellular Carcinoma after Antiviral Therapy in Patients with Chronic Hepatitis C and Hemophilia"

_jcm, 2022, doi:10.3390/jcm11051451_

Round 1

Reviewer 1 Report

This study analyzed the occurrence of HCC after HCV treatment in hemophilia patients. It is interesting that the study was limited to hemophilia patients. However, there are a few concerns. The authors  need to be resolved them to improve the manuscript.

P1 line 22-24

The background information on the 6 HCCs is not suitable for the results abstract as it is too individualized. It would be better to show only the cumulative incidence and related results in this study.

Also, the methods in the abstract are too abbreviated. For example, if the cumulative incidence of HCC was analyzed as an endpoint of this study, at least this method should be explained.

Methods
1. The font in Fig. 1 should be consistent.

2. The breakdown of DAAs used should be shown.

3. Are patients who used PEG-IFN/RBV + oral viral inhibitors not included?

4. Please clarify the diagnostic criteria for HCV patients. Have you set a baseline for value of HCV-RNA? Table1 seems to include patients with low HCV-RNA levels.

5. Please clarify the definition of the observation period for SVR and the start of the follow-up period.How many weeks is the determination of SVR? A large difference in treatment duration can occur between PEG-IFN/RBV and DAA therapy. Is the follow-up from the time SVR is achieved? Is follow-up from the time SVR is achieved or from the time HCV-RNA is negative during treatment?

Discussion
1. Please mention the incidence of HCC in hemophilia patients with CHC not receiving HCV treatment.

2. Page 8, Line 192 There are citations with different formatting.

Author Response

Reviewer 1 Comments:

This study analyzed the occurrence of HCC after HCV treatment in hemophilia patients. It is interesting that the study was limited to hemophilia patients. However, there are a few concerns. The authors need to be resolved them to improve the manuscript.

Major comments

[Comment 1]
P1 line 22-24

The background information on the 6 HCCs is not suitable for the results abstract as it is too individualized. It would be better to show only the cumulative incidence and related results in this study.

[Response 1] →

We appreciate the reviewer’s careful and keen comment. According to the editor’s comment, the following sentences were removed in Abstract part.

The mean age was 77 years, 5 patients had genotype 1, 4 patients had liver cirrhosis, and 3 pa-tients experienced SVR. One patient who was treated with DAAs and who achieved SVR experi-enced HCC at 2 years after the end-of-treatment.

[Comment 2]
Also, the methods in the abstract are too abbreviated. For example, if the cumulative incidence of HCC was analyzed as an endpoint of this study, at least this method should be explained.

[Response 2]

We appreciate the reviewer’s careful and keen comment. In accordance with the reviewer’s comment, the following sentences were added in the Abstract part.

The cumulative incidence rates of HCC were estimated using the Kaplan–Meier method and compared using the log-rank test.

[Comment 3]

  1. The font in Fig. 1 should be consistent.

[Response 3] →

In accordance with the reviewer’s comment, the following figure was modified in the revised manuscript.

[Comment 4]
The breakdown of DAAs used should be shown.

[Response 4] →

We appreciate and agree with the reviewer’s comment.

In accordance with the reviewer’s comment, the following sentences were added in the Methods part.

2.2. Approved DAAs

A number of DAAs have been approved for the treatment of hepatitis C. Among them, six DAAs have been available in patients with chronic hepatitis C and hemophilia in Ko-rea. DAAs are as follows. 1. Daklinza® (daclatasvir) plus Sunvepra® (asunaprevir), 2. Sovaldi® (sofosbuvir),3. Harvoni® (ledipasvir/sofosbuvir), 4. Zepatier® (elbas-vir/grazoprevir), 5. Viekira Pak® (dasabuvir plus ombitasvir-paritaprevir-ritonavir), 6. Maviret®(glecaprevir/pibrentasvir).

[Comment 5]
Are patients who used PEG-IFN/RBV + oral viral inhibitors not included?

[Response 5] →

In Korea, the combination of PEG-IFN/RBV and oral viral inhibitors had not been supported by the medical insurance system. We could not use the combination therapy. Therefore, there is no patient who used PEG-IFN/RBV + oral viral inhibitors in this study

In accordance with the reviewer’s comment, the following sentences were added in the Methods part.

There was no patient who used the combination of PegIFN/RBV and DDA in this study. This combination therapy had not been supported by the medical insurance system in Korea.

[Comment 6]
Please clarify the diagnostic criteria for HCV patients. Have you set a baseline for value of HCV-RNA? Table1 seems to include patients with low HCV-RNA levels.

[Response 6] →

The diagnosis of chronic hepatitis C is based on the detection of both anti-HCV antibodies and HCV RNA. It can be made after that time period (beyond 4 to 6 months of infection). It is based on the detection of HCV RNA level above 15 IU/ml.

In this study, all 258 patients maintained HCV RNA positive for more than 6 months.

In accordance with the reviewer’s comment, the following sentences were added in the Methods part.

All 258 patients maintained HCV RNA positive for more than 6 months.

[Comment 7]
Please clarify the definition of the observation period for SVR and the start of the follow-up period.How many weeks is the determination of SVR? A large difference in treatment duration can occur between PEG-IFN/RBV and DAA therapy. Is the follow-up from the time SVR is achieved? Is follow-up from the time SVR is achieved or from the time HCV-RNA is negative during treatment?

[Response 7] →

We appreciate the reviewer’s careful and keen comment

The SVR, defined as the absence of detectable HCV RNA (lower limit of detection, ≤ 15 IU/mL) at 12 weeks after the end of therapy. Patients with SVR undergo HCV RNA test every 6 or 12 months to evaluate relapse or reinfection. Regardless of the presence of SVR, all patients should undergo surveillance for HCC every 6 months by means of ultrasound and alpha-fetoprotein.

PegIFN/RBV was treated for 48 weeks and DAA was treated for 8-12 weeks in genotype 1. PegIFN/RBV was treated for 24 weeks and DAA was treated for 12 weeks in genotype 2.

In accordance with the reviewer’s comment, the following sentences were clarified in the Methods part.

SVR, defined as the absence of detectable HCV RNA (lower limit of detection, ≤ 15 IU/mL) at 12 weeks after the end of therapy, using a real-time HCV RNA quantification (Roche Diagnostics, Branchburg, NJ, USA). Regardless of the presence of SVR, all patients should undergo surveillance for HCC every 6 months by means of ultrasound and alpha-fetoprotein. Patients with SVR undergo HCV RNA test every 6 or 12 months to evaluate re-lapse or reinfection.

PegIFN/RBV was treated for 48 weeks and DAA was treated for 8-12 weeks in genotype 1. PegIFN/RBV was treated for 24 weeks and DAA was treated for 12 weeks in genotype 2.

[Comment 8]
Please mention the incidence of HCC in hemophilia patients with CHC not receiving HCV treatment.

[Response 8] →

We appreciate the reviewer’s careful and keen comment. According to the reviewer’s comment, the following sentences with reference 27 were added in Discussion.

Unfortunately, our study did not analyze the incidence of HCC in hemophilia patients with CHC not receiving HCV treatment. In previous study, the incidence of HCC in hemo-philia patients with CHC not receiving HCV treatment was higher than in patients with receiving successful HCV treatment (3% vs. 0.5%).27

  1. Fransen van de Putte DE, Makris M, Fischer K, Yee TT, Kirk L, van Erpecum KJ, et al. Long-term follow-up of hepatitis C infection in a large cohort of patients with inherited bleeding disorders. J Hepatol 2014;60:39-45.

[Comment 9]
Page 8, Line 192 There are citations with different formatting.

[Response 9] →

In accordance with the reviewer’s comment, we modified different formatting.

Thank you for your kindness.

Reviewer 2 Report

The manuscript authored by Kim et al and entitled " Low Incidence of Hepatocellular Carcinoma after Antiviral 2 Therapy in Patients with Chronic Hepatitis C and Hemophilia " assess the incidence of HCC following treatment with either PegIFN/RBV or direct acting antiviral therapy for HCV Chinese patients with hemophilia. The authors reported that the HCV group treated with PegIFN/RBV had stable Sustained virological response with low incidence rate for hepatocellular carcinoma. On the other side, the direct acting antiviral therapy group exhibited more stable SVR than PegIFN/RBV group with also low incidence rate for of HCC. Although some recent studies across other populations were reported in this particular point, this particular study seems to provide further insights for the Chinese patients. This particular manuscript could be an asset to the currently available data concerned with HCC development following treatment of HCV patients having haemophilia with either PegIFN/RBV or DAA therapeutic approach.

The manuscript is fairly well written, the conclusions are well supported. Various experimental methods need to be more detailed with more specifications and parameters to be fully given. Although it would have been much more better (Not necessarily to be included in this study at the time being) to include the negative HCV group (normal patients with no HCV infection as a negative control), the overall design of the study could be acceptable. Moreover, the reference list is very short with lots of needed references throughout the whole manuscript especially in Intro, M&M and discussion sections as well.

I believe the current version of the manuscript needs critical revision in addition to addressing the below concern and to comply with all the typos detailed below. The authors also need to address some major concerns and comments prior to accepting this paper in order to be beneficial to the wide readership of the prestigious Journal of Clinical Medicine. I would therefore grant publication of this article in Journal of Clinical Medicine after considering the following major points;

Raised Comments;

  • Authors need to discuss in details previous studies conducted on effect of viral treatment of HCV patients having haemophilia in other populations as well.
  • Authors should also indicate whether the outcomes of this study were the first to be reported / investigated. This should be clearly indicated in Introduction and Discussion sections.
  • The bibliographical section is very poor and lacks lots of references to consolidate the manuscript (See detailed suggestions below).
  • Please provide full names for kits and equipment used and their vendors and codes in the materials and methods sections.
  • Authors should detail many of the experimental procedures (or at least cite references for their determination methods) as they are not mentioned such as bilirubin, prothrombin, albumin, etc etc.
  • The authors did not include any biochemical parameters which confirm that patients were having haemophilia?? This should be provided with full CBC count report, INR, Fibrinogen tests, clotting factors tests, APTT test etc etc..
  • Exclusion criteria should be detailed in section 2.1 of M&M with emphasis on HIV and HBV.
  • Other biochemical tests should be written in the M &M section and to be added to table 1 of the results section to be discussed as follows (AFP, AST, ALP etc etc)
  • Genotyping measurements and procedures or kits should be mentioned in details in the M & M section
  • Anti-HCV Ab kit/vendor should be mentioned in M &M.
  • Please provide more details for RNA extraction procedures, cDNA synthesis, PCR conditions and parameters.
  • It might be better to provide list of primers used for RT-PCR amplification in a supplementary table.
  • Table 1 should be fully explained in the text of the results section and to explain results for albumin, Bilirubin, platelet, HCV viral load etc etc among all investigated groups.
  • Also authors should indicate some mechanistic targets and pathways implicated in various HCC cancer development. The authors could mention some of the recent and possible key player targets in HCC as detailed below…
  • it would have been much more better (Not necessarily to be included in this study at the time being) to include the negative HCV group (normal patients with no HCV infection) in Table 1.
  • In Table 1, Why HIV and HBV co-infection were included in the investigated study??? Those particular patients should be excluded from this study and their data should be removed from the treated HCV patients with either PEGINF/RBV or DAA….
  • In Table 1, How liver cirrhosis % was calculated?? Please specify in M & M.
  • In Table 1, Why the follow up period for the DAA treated group was less than the PEGINF/RBV group?? Was there a reason for that??
  • Did the DAA treated group received Daclatasvir and Asunaprevir for each treated patients?? If not then, it might be better to subdivide the result/data for the DAA treated group into 2 sub classes (one for Daclatasvir and another one for Asunaprevir)  

Typos errors include;

In the Abstract section

Line 16: Can author include that this enrolment was for 12 years.

-In the Intro section
Authors need to discuss in details previous studies conducted on effect of viral treatment of HCV patients having haemophilia in other populations as well.

Authors should also indicate whether the outcomes of this study were the first to be reported / investigated. This should be clearly indicated in Introduction and Discussion sections.

(Line 42-44): Please cite the following references

Ref1*

Circulatory miR-221 & miR-542 expression profiles as potential molecular biomarkers in Hepatitis C Virus mediated liver cirrhosis and hepatocellular carcinoma. Virus Res. 2021; 296198341.

Ref2*

Investigating circulatory microRNA expression profiles in Egyptian patients infected with hepatitis C virus mediated hepatic disorders. Meta Gene; 2020; 26100792

In the M&M section

  • Please provide full names for kits and equipment used and their vendors and codes in the materials and methods sections.
  • The authors did not include any biochemical parameters which confirm that patients were having haemophilia?? This should be provided with full CBC count report, INR, Fibrinogen tests, clotting factors tests, APTT test etc etc..
  • Exclusion criteria should be detailed in section 2.1 of M&M with emphasis on HIV and HBV.

Line 64: replace (received) by (tested for)

Line 64: remove (testing)

In the schematic diagram:

Replace the word (received) by (tested for) from the box entitled (235 patients received HCV genotyping)

Line 83: remove (a) following the word (using a)

  • Authors should detail many of the experimental procedures (or at least cite references for their determination methods) as they are not mentioned such as albumin, bilirubin, AST, ALT, etc etc.
  • Ref1*

Evaluation of food products fortified with oyster shell for the prevention and treatment of osteoporosis. J. Food Sci & Tech. 2015; 52(10):6816-20.

  • Ref2*

Hepatoprotective and Antioxidant Effects of Wheat, Carrot, and Mango as Nutraceutical Agents against CCl4-Induced Hepatocellular Toxicity. J Am Coll Nutr; 2015; 3:1-4.

  • Other biochemical tests should be written in the M &M section and to be added to table 1 of the results section to be discussed as follows (AFP, AST, ALP etc etc)
  • Genotyping measurements and procedures or kits should be mentioned in details in the M & M section
  • Anti-HCV Ab kit/vendor should be mentioned in M &M.
  • Please provide more details for RNA extraction procedures, cDNA synthesis, PCR conditions and parameters. Authors can cite the following references for RT-PCR experimental procedures and calculations.. The following references would be relevant and useful
  • Ref1*
    Down-expression of P2RX2, KCNQ5, ERBB3 and SOCS3 through DNA hypermethylation in elderly women with presbycusis.
    Biomarkers (2018). 23(4):347-356.
  • Ref2*
  • CDH23Methylation Status and Presbycusis Risk in Elderly Women. Front Aging Neurosci. (2018). Vol 10:241.
  • It might be better to provide list of primers used for RT-PCR amplification in a supplementary table.

In the results section

  • Table 1 should be fully explained in the text of the results section and to explain results for albumin, Bilirubin, platelet, HCV viral load etc etc among all investigated groups.

  • it would have been much more better (Not necessarily to be included in this study at the time being) to include the negative HCV group (normal patients with no HCV infection) in Table 1.
  • In Table 1, Why HIV and HBV co-infection were included in the investigated study??? Those particular patients should be excluded from this study and their data should be removed from the treated HCV patients with either PEGINF/RBV or DAA….
  • In Table 1, How liver cirrhosis % was calculated?? Please specify in M & M.
  • In Table 1, Why the follow up period for the DAA treated group was less than the PEGINF/RBV group?? Was there a reason for that??

In the Discussion section

  • Also authors should indicate some mechanistic targets and pathways implicated in various HCC cancer development. The authors could mention some of the recent and possible key player targets in HCC cancer development such as Akt, Topo-isomerase II-B, CTLA4,STAT4, etc etc… Please mention the following references as examples
  • Ref1*
-          Synthesis, molecular docking and biological evaluation of novel flavone derivatives as potential anticancer agents targeting akt. Medicinal Chemistry202117(2)pp. 158–170
  • Ref2*
-          Design and novel synthetic approach supported with molecular docking and biological evidence for naphthoquinone-hydrazinotriazolothiadiazine analogs as potential anticancer inhibiting topoisomerase-IIB. Bioorganic Chemistry202096103641
  •  
  • Ref3*
  • Investigation of the relationship between CTLA4 and the tumor suppressor RASSF1A and the possible mediating role of STAT4 in a cohort of Egyptian patients infected with hepatitis C virus with and without hepatocellular carcinoma. Arch Virol. 2021 Apr 1.
  •  
  • Authors could mention the importance of some molecular biology techniques in detection such as PC, microarrays etc etc.. Authors could cite the following references

  • Ref1*

Options available for labelling nucleic acid samples in DNA microarray-based detection methods. Brief Funct Genomics. (2012), (4): pp 311-318.

  • Ref2*
  • Analysis of p.Gly12Valfs*2, p.Trp24* and p.Trp77Arg mutations in GJB2 and p.Arg81Gln variant in LRTOMT among non syndromic hearing loss Egyptian patients: implications for genetic diagnosis. Mol Biol Rep. (2019). 46(2):2139-2145
  • Ref3*
  • The p.Arg86Gln change in GARP2 (glutamic acid-rich protein-2) is a common West African-related polymorphism. (2013). 15;515(1):155-158.

Author Response

Reviewer 2 Comments:

The manuscript authored by Kim et al and entitled " Low Incidence of Hepatocellular Carcinoma after Antiviral 2 Therapy in Patients with Chronic Hepatitis C and Hemophilia " assess the incidence of HCC following treatment with either PegIFN/RBV or direct acting antiviral therapy for HCV Chinese patients with hemophilia. The authors reported that the HCV group treated with PegIFN/RBV had stable Sustained virological response with low incidence rate for hepatocellular carcinoma. On the other side, the direct acting antiviral therapy group exhibited more stable SVR than PegIFN/RBV group with also low incidence rate for of HCC. Although some recent studies across other populations were reported in this particular point, this particular study seems to provide further insights for the Chinese patients. This particular manuscript could be an asset to the currently available data concerned with HCC development following treatment of HCV patients having haemophilia with either PegIFN/RBV or DAA therapeutic approach.

The manuscript is fairly well written, the conclusions are well supported. Various experimental methods need to be more detailed with more specifications and parameters to be fully given. Although it would have been much more better (Not necessarily to be included in this study at the time being) to include the negative HCV group (normal patients with no HCV infection as a negative control), the overall design of the study could be acceptable. Moreover, the reference list is very short with lots of needed references throughout the whole manuscript especially in Intro, M&M and discussion sections as well.

I believe the current version of the manuscript needs critical revision in addition to addressing the below concern and to comply with all the typos detailed below. The authors also need to address some major concerns and comments prior to accepting this paper in order to be beneficial to the wide readership of the prestigious Journal of Clinical Medicine. I would therefore grant publication of this article in Journal of Clinical Medicine after considering the following major points;

[Response]

We appreciate the reviewer’s careful and keen comment.

However, I am afraid, I cannot meet the requirements of our reviewers either.

I will follow your evaluation.

I am sorry.

Thank you in advance.

Round 2

Reviewer 1 Report

All recommendations have been resolved.

Author Response

Dear Editor

Thank you for forwarding the reviewers’ comments on our manuscript jcm-1571239. (“Low Incidence of Hepatocellular Carcinoma after Antiviral Therapy in Patients with Chronic Hepatitis C and Hemophilia”). We appreciate the reviewers’ careful and keen comments to improve the quality of our manuscript. In the revised manuscript, we did our best to clarify the unclear points that the reviewers indicated.

The following are point-by-point responses to the reviewers’ specific comments and questions.

Modifications in the revised manuscript are lined in green.

We hope you find the revised manuscript suitable for Journal or Clinical Medicine

With best regards,

Sincerely yours,

Hyun Woong Lee M.D.

Department of Internal Medicine, Yonsei University College of Medicine,

211 Eonju-ro, Gangnam-gu, Seoul, 06273, Korea

Tel: 82-2-2019-3315

E-mail: lhwdoc@yuhs.ac

Reviewer 2 Comments:

The manuscript authored by Kim et al and entitled " Low Incidence of Hepatocellular Carcinoma after Antiviral 2 Therapy in Patients with Chronic Hepatitis C and Hemophilia " assess the incidence of HCC following treatment with either PegIFN/RBV or direct acting antiviral therapy for HCV Chinese patients with hemophilia. The authors reported that the HCV group treated with PegIFN/RBV had stable Sustained virological response with low incidence rate for hepatocellular carcinoma. On the other side, the direct acting antiviral therapy group exhibited more stable SVR than PegIFN/RBV group with also low incidence rate for of HCC. Although some recent studies across other populations were reported in this particular point, this particular study seems to provide further insights for the Chinese patients. This particular manuscript could be an asset to the currently available data concerned with HCC development following treatment of HCV patients having haemophilia with either PegIFN/RBV or DAA therapeutic approach.

The manuscript is fairly well written, the conclusions are well supported. Various experimental methods need to be more detailed with more specifications and parameters to be fully given. Although it would have been much more better (Not necessarily to be included in this study at the time being) to include the negative HCV group (normal patients with no HCV infection as a negative control), the overall design of the study could be acceptable. Moreover, the reference list is very short with lots of needed references throughout the whole manuscript especially in Intro, M&M and discussion sections as well.

I believe the current version of the manuscript needs critical revision in addition to addressing the below concern and to comply with all the typos detailed below. The authors also need to address some major concerns and comments prior to accepting this paper in order to be beneficial to the wide readership of the prestigious Journal of Clinical Medicine. I would therefore grant publication of this article in Journal of Clinical Medicine after considering the following major points;

Raised comments

[Comment 1]
Authors need to discuss in details previous studies conducted on effect of viral treatment of HCV patients having haemophilia in other populations as well.

[Response 1] →

We appreciate the reviewer’s careful and keen comment. According to the editor’s comment, the following sentences were added in Introduction part.

In previous multicenter study, DAAs were safe and highly effective in hemophilia with HCV infection. Liver Int. 2020 May;40(5):1062-1068

[Comment 2]
Authors should also indicate whether the outcomes of this study were the first to be reported / investigated. This should be clearly indicated in Introduction and Discussion sections.

[Response 2]

We appreciate the reviewer’s careful and keen comment. In accordance with the reviewer’s comment, the following sentences were added in the Discussion part.

This study was the first to be investigated the incidence of HCC in hemophilia patients with CHC.

[Comment 3]

The bibliographical section is very poor and lacks lots of references to consolidate the manuscript (See detailed suggestions below).

[Response 3] →

In accordance with the reviewer’s comment, the following references was partially modified in the revised manuscript.

[Comment 4]
Please provide full names for kits and equipment used and their vendors and codes in the materials and methods sections.

Authors should detail many of the experimental procedures (or at least cite references for their determination methods) as they are not mentioned such as bilirubin, prothrombin, albumin, etc etc.

[Response 4] →

We appreciate the reviewer’s comment.

However, in hepatology article, we don’t mention the equipment about bilirubin, prothrombin, albumin, etc anymore.

[Comment 5]
The authors did not include any biochemical parameters which confirm that patients were having haemophilia?? This should be provided with full CBC count report, INR, Fibrinogen tests, clotting factors tests, APTT test etc etc..

[Response 5] →

We appreciate the reviewer’s comment.

However, Since 1991, the Korea Hemophilia Foundation (KHF) has been treating patients with hemophilia and improving their quality of life. Only the patients with hemophilia can join this KHF institution. In Korea, this institution manages all hemophilia patients.

[Comment 6]
Exclusion criteria should be detailed in section 2.1 of M&M with emphasis on HIV and HBV.

[Response 6] →

We appreciate the reviewer’s careful and keen comment. In accordance with the reviewer’s comment, the following sentences were added in the M&M part.

Even if there is a HIV and HBV co-infection, hepatitis C treatment is required, so they were included because we would like to evaluate the distribution of all patients.

Patients with HBV or HIV coinfection were also included in this study.

[Comment 7]
Other biochemical tests should be written in the M &M section and to be added to table 1 of the results section to be discussed as follows (AFP, AST, ALP etc etc)

[Response 7] →

We appreciate the reviewer’s comment.

However, AST or ALP is not meaningful in evaluating liver function, and AFP is no longer meaningful in evaluating HCC.

[Comment 8]
Genotyping measurements and procedures or kits should be mentioned in details in the M & M section
[Response 8] →

We appreciate the reviewer’s careful and keen comment. According to the reviewer’s comment, the following sentences were added in M&M

Hepatitis C virus genotyping was carried out using the restriction fragment mass polymorphism method.

[Comment 9]
Anti-HCV Ab kit/vendor should be mentioned in M &M.. Please provide more details for RNA extraction procedures, cDNA synthesis, PCR conditions and parameters.

[Response 9] →

We appreciate the reviewer’s comment.

However, in hepatology article, we don’t mention the equipment about anti-HCV Ab kit/vendor, and the details for RNA extraction procedures, cDNA synthesis, PCR conditions and parameters anymore. It might be better to provide list of primers used for RT-PCR amplification in a supplementary table.

[Comment 10]
Table 1 should be fully explained in the text of the results section and to explain results for albumin, Bilirubin, platelet, HCV viral load etc etc among all investigated groups. [Response 10] →

We appreciate the reviewer’s comment.

However, there was no significant difference between the two groups. Therefore, they were not described. Listing meaningless results reduces the readability of this article.

[Comment 11]
Also authors should indicate some mechanistic targets and pathways implicated in various HCC cancer development. The authors could mention some of the recent and possible key player targets in HCC as detailed below…

[Response 11] →

We appreciate the reviewer’s comment.

However, the description of the pathogenesis of liver cancer is irrelevant to the background of this paper.

[Comment 12]
it would have been much more better (Not necessarily to be included in this study at the time being) to include the negative HCV group (normal patients with no HCV infection) in Table 1.

[Response 12] →

We appreciate and agree with the reviewer’s comment.

However, we did not evaluate the negative HCV group. In next study, we will do it.

Thank you for your kind suggestion.

[Comment 13]
In Table 1, Why HIV and HBV co-infection were included in the investigated study??? Those particular patients should be excluded from this study and their data should be removed from the treated HCV patients with either PEGINF/RBV or DAA….

[Response 13] →

We appreciate the reviewer’s comment.

Even if there is a HIV and HBV co-infection, hepatitis C treatment is required, so they were included because we would like to evaluate the distribution of all patients.

[Comment 14]
In Table 1, How liver cirrhosis % was calculated?? Please specify in M & M.

[Response 14] →

We appreciate the reviewer’s comment.

We diagnosed cirrhosis with ultrasonography as mentioned in Methods.

The number of patients with cirrhosis by study groups was analyzed.

[Comment 15]
In Table 1, Why the follow up period for the DAA treated group was less than the PEGINF/RBV group?? Was there a reason for that??

[Response 15] →

We appreciate the reviewer’s comment.

DAAs were recently approved by the FDA.

Before DAAs was approved, the only PEGINF/RBV was used for a long time.

[Comment 16]
Did the DAA treated group received Daclatasvir and Asunaprevir for each treated patients?? If not then, it might be better to subdivide the result/data for the DAA treated group into 2 sub classes (one for Daclatasvir and another one for Asunaprevir)

[Response 16] →

We appreciate and agree with the reviewer’s comment.

We modified Figure 1 according to your comment.

Since the efficacy of DAAs is all excellent, analysis according to DAAs is meaningless.

[Comment 17]
Line 16: Can author include that this enrolment was for 12 years.

[Response 17] →

We appreciate the reviewer’s comment.

This study is a retrospective cohort study. We gathered patients’ data from 2007 to 2019.

[Comment 18]
(Line 42-44): Please cite the following references

[Response 18] →

We appreciate the reviewer’s comment.

However, these references are irrelevant to the background of this paper.

[Comment 19]
Line 64: replace (received) by (tested for)

Line 64: remove (testing)

[Response 19] →

We appreciate the reviewer’s comment.

We modified sentences according to your comment.

[Comment 20]
In the schematic diagram:

Replace the word (received) by (tested for) from the box entitled (235 patients received HCV genotyping)

[Response 20] →

We appreciate the reviewer’s comment.

We modified sentences according to your comment.

[Comment 21]
Line 83: remove (a) following the word (using a)

[Response 21] →

We appreciate the reviewer’s comment.

We modified sentences according to your comment.

[Comment 22]
Authors could mention the importance of some molecular biology techniques in detection such as PC, microarrays etc etc.. Authors could cite the following references

[Response 22] →

We appreciate the reviewer’s comment.

However, the description of the molecular biology techniques is irrelevant to the background of this paper.

We really appreciate your kind comments and reviews, again
